# Climate Crisis and Mental Well-Being: Nature Relatedness, Meaning in Life, and Gender Differences in a Jewish Australian Study

**DOI:** 10.3390/bs15081045

**Published:** 2025-08-01

**Authors:** Orly Sarid

**Affiliations:** The Spitzer Department of Social Work, Goldman Sonnenfeldt School of Sustainability and Climate Change, Ben-Gurion University of the Negev, Beer-Sheva 84105, Israel; orlysa@bgu.ac.il

**Keywords:** climate change, meaning in life, nature relatedness, gender differences, Jewish traditions, mental health

## Abstract

Background: Amid growing concerns about climate crisis and its psychological toll, understanding how people find meaning through their connection to nature is increasingly important. The first aim of this study is to examine the association between Nature Relatedness (NR) and Meaning in Life (MIL). The second aim is to investigate if gender moderates this association and to explore how Jewish traditions influence gender differences in this relationship. Methods: A multi-methods design was employed. Participants were recruited through purposive sampling of prominent Jewish community figures, followed by snowball sampling via informant referrals. Thirty-five participants completed the Meaning in Life Questionnaire (MLQ) and the NR Scale. Two questions provided qualitative insights into participants’ personal interpretations and culturally grounded meanings of NR and MIL in the context of climate change and Jewish traditions. Results: Hierarchical multiple regression analyses assessed the main effects of NR and gender, as well as their interaction, on MLQ subscales. NR positively correlated with the MLQ Search dimension, indicating that individuals with stronger NR actively seek meaning in life. Gender moderated this relationship: NR did not correlate with MLQ Presence overall, but higher NR was linked to greater MIL presence among female participants. Thematic analysis of qualitative responses revealed gender-based variations and emphasized the role of Jewish teachings in connecting NR to cultural and religious practices. Conclusions: The findings point to the importance of cultural, religious, and gender factors in shaping the relationship between NR and MIL in a time of climate change crisis, offering implications for positive mental health research and culturally sensitive interventions.

## 1. Introduction

Climate change is one of the most significant environmental crises of the 21st century, posing profound challenges to individuals, communities, and ecosystems worldwide. Beyond its physical manifestations—such as rising temperatures, sea-level rise, and extreme weather events—climate change is increasingly recognized as a psychological and existential stressor that can disrupt individuals’ sense of safety, continuity, and meaning ([34]).

Australia provides a compelling case study of these dynamics. In 2023, the country shifted from La Niña to El Niño conditions, bringing reduced rainfall, higher temperatures, and an unusually early bushfire season in the eastern states ([30]). These climatic changes have triggered cascading effects on biodiversity, agriculture, and public health, which may in turn contribute to psychological distress—including eco-anxiety and climate-related trauma ([11]).

Approximately 100,000 Australians identify as Jewish—about 0.4% of the national population—with most living in Victoria and New South Wales and smaller communities spread across every other state ([4]). Grounded in teachings that stress stewardship of the Earth and communal responsibility, these communities cultivate an ecological ethos that aligns closely with nature-focused frameworks of meaning ([41]).

In this study, we examine the links between Meaning in Life (MIL) and Nature Relatedness (NR) in the context of climate change in Australia. We acknowledge Jewish traditions and practices that support NR and MIL and may serve as protective factors during times of environmental stress.

### 1.1. Literature Review

Meaning in Life (MIL) addresses existential needs such as purpose, fulfilment, and a sense of coherence ([38]). It is shaped by individuals’ subjective perceptions and reflects their lived experiences, as they endeavor to cultivate a coherent and purposeful sense of existence. Drawing on Antonovsky’s conceptualization ([1]), [27] ([27]) highlight three core elements in MIL: comprehension (making sense of life), purpose (pursuing valued goals), and existential mattering (believing one’s life matters), which collectively influence one’s experience of meaning and are recognized as dimensions of positive mental health. To maintain clarity and consistency, in this paper, we adopt the single term “positive mental health” to denote optimal psychological functioning and subjective wellbeing ([26]). MIL can be measured using the Meaning in Life Questionnaire (MLQ), which comprises two subscales: Presence of Meaning and Search for Meaning ([40]). A recent meta-analysis indicated that perceiving a presence of meaning in life is associated with lower depressive symptoms, whereas actively searching for meaning is linked to higher levels of depression ([21]). This bidirectional pattern can be interpreted through the lens of self-regulation theory: having a sense of meaning supports a coherent identity narrative that buffers against mood dysregulation. In contrast, high search scores may reflect ongoing existential vulnerability and an increased risk for depression. A review of empirical studies confirmed that enhanced emotion-regulation capacity and goal-directed thinking are among the most reliable cognitive–affective mechanisms through which purpose in life promotes positive mental health ([14]).

Close relationships and a sense of belonging can also contribute to a higher sense of meaning by providing structure, purpose, and the understanding that one matters to others. In this regard, participating in a congregation or engaging in activities and ceremonies within a religious community can offer a supportive network with shared values and perspectives. Within a religious system, MIL can be reinforced through structured beliefs and practices, thereby helping individuals maintain a sense of purpose and significance even in the face of adversity.

Nature Relatedness (NR), conceptualized as an individual’s identification with and connection to the natural environment, is a multidimensional construct that encompasses the affective, cognitive, and experiential dimensions of one’s relationship with nature ([32]). This sense of connectedness emphasizes the individual’s role within the larger ecological system and fosters self-discovery and personal growth as one reflects on their place in the natural world.

Previous research has demonstrated that individuals with higher levels of NR tend to experience greater life satisfaction, report more frequent positive emotions, and engage in an increased number of pro-environmental behaviors. These findings suggest that NR not only enhances positive mental health but also encourages a commitment to sustainability and environmentally responsible practices ([10]; [18]; [33]; [37]; [44]).

Encounters with awe-inspiring natural environments often elicit feelings of wonder, which can foster a connection to something greater than oneself and may evoke humility, ultimately heightening one’s sense of meaning ([27]). Nature’s inherent cyclical rhythms—such as day and night cycles, changing seasons, and growth patterns—provide a coherent framework that helps people feel the world makes sense. However, contemporary climate changes introduce elements of unpredictability, stress, and disruption, potentially undermining this sense of coherence and impairing the experience of meaning ([17]). These dynamics highlight the necessity of examining how a sense of connection to the natural world is linked to environmental engagement and the construction of personal meaning.

### 1.2. Theoretical Framework

The theoretical framework linking NR to MIL can be examined through the Biophilia Hypothesis, which suggests that humans possess an innate affinity for nature, supporting positive mental health and fostering a sense of belonging within the broader ecological system ([45]). These natural connections help fulfil intrinsic psychological needs such as relatedness, autonomy, and purpose. Complementing these psychological theories, religious identity adds depth to our understanding of how NR influences MIL by shaping individuals’ sense of purpose through cultural practices and community involvement that contribute to the meaning-making process ([22], [23]).

Adding an ecological systems perspective further enriches the model by situating NR and MIL within the nested social–environmental contexts that shape individual experience. Drawing on Bronfenbrenner’s Ecological Systems Theory ([9]), five concentric levels can be integrated with Social Role Theory ([13]) to explain how gendered patterns of NR and MIL emerge across cultural and community contexts.

At the microsystem level, day-to-day interactions with family, peers, and immediate natural settings can enhance early affective bonds with nature. Because girls and boys often receive different expectations for caretaking or exploration, gendered trajectories of NR may begin in early childhood ([13]). This aligns with the Biophilia Hypothesis by suggesting that such early experiences can activate innate tendencies toward nature affiliation, which in turn support existential meaning.

The mesosystem—interconnections among home, school, and community institutions—can reinforce or weaken NR through coordinated or conflicting messages. For instance, educational and youth programming may amplify relational values traditionally socialized in girls, thereby deepening the NR-MIL pathway for women. When framed through culturally salient themes such as responsibility and continuity, these same processes may also enhance this pathway for men.

The exosystem includes institutions and social forces that influence individuals indirectly, such as environmental policy, media narratives, and organizational structures. These may shape opportunities for environmental engagement that appeal differently to men and women depending on prevailing social roles.

The macrosystem—encompassing broad cultural ideologies and belief systems—supplies value frameworks and moral norms that legitimize environmental attitudes and behaviors as meaningful. Social Role Theory posits that these narratives interact with gender norms, whereby women may frame environmental concern through a relational and care-oriented lens, while men may emphasize duty and moral leadership. Integrating Relational-Cultural Theory ([24]) strengthens this account. If nature is perceived as a relational entity, women may engage with it in ways that mirror their approach to human relationships, enhancing the NR–MIL connection.

Finally, the chronosystem captures socio-historical change. Accelerating environmental disruptions and evolving social expectations alter the way individuals relate to nature over time, prompting shifts in how NR and MIL are experienced across generations ([17]).

### 1.3. Research Aims and Hypotheses

The first aim of the study is to explore, in times of major climate change, the association between NR and MIL. The second aim is to investigate whether gender moderates this association and to explore whether Jewish traditions influence gender differences in this relationship within the context of climate change.

We hypothesize that: 

**Hypothesis 1.** 
*There will be a positive association between NR and MIL.*


**Hypothesis 2.** 
*Gender will moderate the association between NR and MIL.*


**Hypothesis 3.** 
*Jewish traditions will enhance the link between NR and MIL, reflecting culturally embedded pathways to meaning even in times of climate change.*


## 2. Method

### 2.1. Study Design

This cross-sectional study used an explanatory sequential mixed-methods design. A quantitative survey was conducted first to provide breadth, followed by two open-ended qualitative questions to add depth. The design was partially integrated, as defined by [28] ([28]). Integration was performed at the interpretation stage: quantitative patterns were revisited alongside qualitative themes and were found to point in the same direction, so we drew one integrated conclusion, following the approach described by [36] ([36]).

### 2.2. Ethical Considerations

Ethical approval was granted by the Spitzer Department of Social Work Ethics Committee, Ben-Gurion University of the Negev (#OT05052024). All procedures adhered to the Declaration of Helsinki. Participation was voluntary: every participant gave written informed consent and could withdraw at any time, although none did so. Questionnaires were coded to preserve anonymity, and all data were self-reported.

### 2.3. Setting and Participants

#### Setting

The study was conducted in Australia between November and December 2023 and included individuals aged 21 and older who self-identified as members of the Jewish community.

### 2.4. Eligibility Criteria

Participants were aged 21 years and older. Although the legal age of adulthood in Australia is 18, developmental research positions ages 18 to 25 as “emerging adulthood”, a phase of ongoing neurocognitive maturation and variable dependency ([2]). National projections show that about one-half of Australians aged 20 to 24 still reside in the parental home, whereas roughly one-fifth of those aged 25 to 29 do so ([5]). A 21-year threshold therefore excludes much of the most dependent cohort while retaining adults whose household, transport, and civic behaviors are more likely to be self-determined.

### 2.5. Recruitment and Sampling

#### Sampling Strategy

Recruitment was conducted in two stages. In the first stage, purposive sampling was used to identify 26 prominent Jewish figures—such as rabbis, lay leaders, and educators—through community websites, synagogue newsletters, professional directories, and social media groups. These individuals served as key informants. In the second stage, snowball sampling was employed, with each informant referring to two additional eligible contacts. Snowball sampling is considered appropriate when the target population is network-bound, geographically dispersed, and not easily accessible through public membership lists ([3]; [35]). This recruitment strategy mirrors the method described by [39] ([39]), who engaged ultra-Orthodox mothers in Israel through community leaders and participant referrals.

### 2.6. Participant Characteristics

The study included 35 Jewish participants, aged between 22 and 85 years (M = 60.49, SD = 16.42). The majority were women (n = 21, 60.0%), in a relationship (n = 25, 71.4%), and reported having children (n = 29, 82.9%). In terms of economic status, 54.3% (n = 19) indicated ‘doing very well’, while 45.7% (n = 16) reported ‘doing well’. Years of education ranged from 12 to 25 years (M = 19.54, SD = 3.66). Most participants were born in Australia (n = 21, 60.0%), while those born elsewhere had lived in Australia for 22 to 75 years (M = 43.87, SD = 16.40). Regarding religious affiliation, 66% (n = 23) identified as active or partly active in the Progressive Jewish Movement, 10 reported no affiliation with any specific denomination, and two identified as Orthodox.

### 2.7. Sample Size Justification

To determine the required sample size for testing the moderation hypothesis, a power analysis was conducted using G*Power 3.1.9. ([15]). This analysis aimed to detect a large effect size (f^2^ = 0.35) with at least 80% power (1-β) and an alpha level of 0.05, estimating that a sample size of 25 would be sufficient. The actual sample size of 35 participants exceeded this requirement, thereby ensuring more than adequate statistical power for the analysis

### 2.8. Data Collection Procedures

Participants were recruited through email invitations. After participants were approached, the aims of the study were explained to them. Those who agreed to take part signed an informed consent form and completed a digital questionnaire, which took approximately 20 min to complete. Of the 37 persons approached, two declined due to scheduling constraints. Twenty-three participants provided narrative responses to the open-ended questions (qualitative strand).

### 2.9. Measures

#### Quantitative Measures

The *Nature Relatedness* (NR) Scale is a 21-item scale designed to measure individuals’ affective, cognitive, and experiential relationships with the natural environment ([32]). The scale measures the extent to which individuals internalize their connection to nature. Each item is rated on a 5-point Likert scale ranging from 1 (disagree strongly) to 5 (agree strongly). Sample items include “I always think about how my actions affect the environment” and “I feel very connected to all living things and the earth”. The NR scale comprises three subscales: NR-Self, reflecting an internalized identification with nature; NR-Perspective, representing an external, ecological perspective; and NR-Experience, capturing physical familiarity with the natural world. The NR Scale has demonstrated strong psychometric properties, including high internal consistency (Cronbach’s alpha = 0.87) and construct validity ([32]). In this study, an overall NR score was calculated by averaging responses across all items, with higher scores indicating a stronger connection to nature. Cronbach’s alpha = 0.91.

The *Meaning in Life Questionnaire (MLQ)* is a 10-item self-report instrument designed to measure life meaning ([40]). The MLQ consists of two subscales: Presence of Meaning, which assesses the degree to which individuals perceive their lives as meaningful, and Search for Meaning, which evaluates the extent to which individuals are actively seeking meaning in their lives. Responses are recorded on a 7-point Likert scale ranging from 1 (Absolutely Untrue) to 7 (Absolutely True). The MLQ has demonstrated good internal consistency, with coefficient alphas ranging from the low to high 0.80 s for the Presence subscale and mid 0.80 s to low 0.90 s for the Search subscale. In the current study, MLQ Search Cronbach’s alpha = 0.90; MLQ Presence Cronbach’s alpha = 0.83.

*Socio-Demographic data* included gender, age, place of birth, year of immigration, family status, having children, education, and economic status.

### 2.10. Qualitative Measures

The qualitative component of the study consisted of two questions designed to elicit personal reflections on the role of Jewish teachings and cultural practices in shaping attitudes toward nature and meaning in life during times of climate change:Are there any Jewish traditions or cultural practices that have been particularly helpful in shaping your attitudes toward nature in the context of climate change?Are there any Jewish traditions or cultural practices that have been particularly helpful in supporting your search for meaning in life during times of climate change?

### 2.11. Data Analysis

#### 2.11.1. Quantitative Analysis

Quantitative data analyses were performed using IBM SPSS Statistics version 29. Results were deemed statistically significant when *p* < 0.05, while marginal significance was reported at *p* < 0.10.

Initially, the data were screened for outliers and assessed for distribution. No outliers were detected ([19]), and all variables exhibited normal skewness and kurtosis. Subsequently, intercorrelations between NR and MLQ measures, along with their associations with background variables, were analyzed. Continuous variable associations were examined using zero-order correlations, while point-biserial correlations were employed for dichotomous and continuous variable associations.

#### 2.11.2. Moderation Analysis

Moderation analyses were conducted using hierarchical multiple regression models in two steps. In Step 1, gender and nature relatedness were included, while Step 2 involved the addition of the interaction term between these two variables. The MLQ measures served as dependent variables. NR was centered prior to analysis to improve interpretability. The interaction was then probed and visualized using the simple slopes method via the PROCESS macro for SPSS (Model 1) ([20]). Variance inflation factor (VIF) values were checked to ensure multicollinearity was not an issue (i.e., VIF > 10).

#### 2.11.3. Qualitative Analysis

The qualitative component consisted of two questions. Responses were analyzed following Braun and Clarke’s framework ([7], [8]).

Phase 1: Two doctoral level social work researchers, both embedded in Jewish cultural contexts, read each of the 23 verbatim transcripts twice—first for holistic understanding and reflexive memo writing, then to highlight meaning-rich segments. Early notes captured, for example, RA (53)’s claim that humans were “created to be caretakers and gardeners of the earth” and PD (74)’s observation that “our holidays are tied to the seasons”, signaling ritual–nature connections.

Phase 2: Working independently, each researcher produced short, semantic data-driven codes for every marked segment, generating 68 unique codes such as “ritualised stewardship”, “ethical stewardship”, and “relational care for animals”. PB (78)’s description of Bal Tashchit as “a meaningful cultural method to care for the planet” received both “ritualised stewardship” and “symbolic action”.

Phase 3: Through collaborative reflexive discussion, overlapping codes were grouped into nine provisional clusters. For example, codes on “ethical stewardship”, “Torah mandates”, and “moral clarity” converged into “Jewish ethics”.

Phase 4: Clusters were iteratively compared with the full data set, collapsing into three coherent candidate themes that captured shared meanings across participants and genders, while retaining deviant cases like VR (70)’s comment that he “does not explicitly link” heritage to activism.

Phase 5: We decided on three final theme names: “*Jewish Teachings as Ethical Foundations for NR and the Presence of Meaning*”; “*Rituals and Holidays as Opportunities for Ecological Awareness and Meaning-Making*” and “*Interconnectedness and Reflection as Expressions of NR and the Search for Meaning*”. To explore gender-based differences, responses were analyzed separately for male and female participants.

Phase 6: Themes were written up in a narrative format using quotes to reflect the depth and diversity of participant meaning-making. A detailed audit trail, ongoing peer debriefing, and two rounds of participant checking enhanced the credibility and transparency of the analysis, in keeping with Braun and Clarke’s emphasis on reflexivity and methodological integrity in thematic interpretation.

## 3. Results

### 3.1. Quantitative Findings

Table 1 displays the descriptive statistics and intercorrelations between NR and MLQ measures, as well as their associations with background variables. NR was positively and significantly associated with MLQ Search, but not with MLQ Presence. Regarding background variables, a marginally significant association was found between gender and NR, with female participants scoring higher than males. See Appendix A for means and standard deviations of NR and MLQ measures by dichotomous background variables.

Table 2 presents the results of the hierarchical regression analysis. In both models, multicollinearity was not a concern (all VIF values < 2.14). In the model predicting MLQ Search, Step 1 was marginally significant, with NR positively predicting MLQ Search. However, the inclusion of the interaction term in Step 2 did not yield significant results.

In contrast, in the model predicting MLQ Presence, Step 1 was non-significant, but the addition of the interaction term in Step 2 was significant. As illustrated in Figure 1, a simple slopes analysis revealed that the association between NR and MLQ Presence was positive and significant for women, but non-significant for men. Therefore, Hypothesis 2 was partially supported.

### 3.2. Qualitative Findings

A total of 23 participants (11 men and 12 women) provided qualitative responses.

Three themes emerged from Table 3. that reflect how participants relate Jewish traditions to NR values and MIL—both presence and search—within the context of climate change. While these themes were shared across genders, notable gender differences were also observed. The themes are presented below:


*Jewish Teachings as Ethical Foundations for NR and the Presence of Meaning*


This theme was expressed by 8 of 11 male participants (73%) and 6 of 12 female participants (50%). Male participants frequently cited Jewish teachings—particularly Tikkun Olam, Torah commandments, and ethical stewardship—as theological justifications for environmental responsibility. These teachings were often framed as affirming a sense of moral clarity and purpose, reflecting a strong presence of meaning.

RA (53): “The Torah teaches that humanity was created to be caretakers/gardeners of the earth… Various mitzvot… demonstrate the Jewish mandate to care for all of God’s creations”.

DA (69): “Jewish teachings are consistent with and support my principles and practices, such as Tikkun Olam”.

Female participants also referenced Jewish ethics, though their reflections were more relational and grounded in care for others, including animals and future generations.

MA (65): “Humanity shares the Earth with countless other organisms and must act as good stewards”.

JT (59): “We must look after those that cannot look after themselves”.

These findings suggest that while both genders drew on Jewish ethics, men emphasized theological grounding and moral clarity, whereas women emphasized relational ethics and care.

2.
*Rituals and Holidays as Opportunities for Ecological Awareness and Meaning-Making*


This theme was identified in 4 male participants (36%) and 3 female participants (25%). Male participants highlighted specific rituals—such as Bal Tashlich, Amidah prayer, and Shabbat—as structured opportunities to reflect on nature and human responsibility. These practices were described as reinforcing a stable worldview and spiritual connection to the environment, reflecting presence of meaning.

PB (78): “The practice of ‘Bal Tashlich’… is a meaningful cultural method to care for the planet”.

DK (76): “The Amidah prayer… acknowledges the importance of sun, rain, dew…”

Female participants emphasized the symbolic and seasonal aspects of holidays, which served as reminders of ecological cycles and human responsibility. These reflections often pointed toward search for meaning, as participants described how traditions help them remain mindful and engaged.

PD (74): “Many Jewish holidays are tied to the seasons, agriculture, and the land… These traditions remind me to remain vigilant in protecting resources”.

Thus, while men emphasized ritual as a source of spiritual grounding, women used ritual for reflection and ecological awareness.

3.
*Interconnectedness and Reflection as Expressions of NR and the Search for Meaning*


This theme was expressed by 3 male participants (27%) and 6 female participants (50%). Male participants who reflected on this theme often did so through philosophical or spiritual lenses, linking Jewish teachings to broader ecological worldviews.

CE (77): “The fundamental core of faith is connected to the concept that we are part of nature… Judaism teaches this”.

FP (58): “Jewish teachings stress respect for nature, other humans, and animals”.

Female participants were more likely to describe a sense of ecological interconnectedness and relational awareness as central to their worldview, engagement with nature, and community.

YA (63) and ZB (60): “Jewish traditions point to the interconnectedness of the world, and thus the need to be aware of other species and the flow of the world”.

CB (40): “Jewish teachings can foster greater environmental education and informed decision-making about climate issues”.

These findings suggest that more women expressed a search for meaning through relational and ecological awareness, while men framed interconnectedness through abstract or theological reflection.

### 3.3. Integrative Interpretation

When examined side by side, the quantitative patterns and qualitative themes told the same story. Higher NR related to active searching for meaning (quantitative), and participants, especially women, described relational, stewardship-based practices and reflections that echoed this search (qualitative). Men more often grounded their statements in theological or ethical certainty, which paralleled the quantitative finding that NR related to the presence of meaning primarily among women. This convergence supports the overall conclusion that Jewish teachings, rituals, and concepts of interconnectedness help shape both nature-related values and meaning-making processes in the context of climate change.

## 4. Discussion

This study investigated the association between NR and MIL among Jewish Australians, with particular attention to gender differences. Employing a multi-methods design, we also explored how Jewish teachings and traditions shape environmental attitudes and perceptions of meaning in the context of the climate crisis.

Quantitative findings revealed a significant positive association between NR and the search for meaning, but not with the presence of meaning. This suggests that individuals who feel more connected to nature may be more actively engaged in existential exploration, rather than experiencing a stable sense of purpose. These findings align with [16]’s ([16]) conceptualization of meaning as a dynamic process of striving and growth. The biophilia hypothesis ([45]) and Stress Recovery Theory ([43]) further support the idea that nature fosters emotional restoration and cognitive clarity. As environmental stress increases, particularly in the context of climate change, individuals may become more attuned to the natural world as a source of insight, prompting deeper reflection on life’s purpose and significance.

However, the lack of a significant relationship between NR and the presence of meaning overall suggests that a connection to nature alone may not be sufficient to generate a stable sense of meaning. Rather, it may serve as a catalyst for ongoing meaning-making. In times of ecological disruption, this process may be especially important, as individuals seek to re-establish coherence and purpose in a rapidly changing world.

Gender emerged as a meaningful moderator in the relationship between NR and meaning. Although gender was not significantly associated with MLQ scores overall, women reported marginally higher levels of NR than men. More importantly, NR was linked to the presence of meaning for women, but not for men. This gender-specific effect may reflect differing socialization processes or varying ways in which men and women interact with and derive value from natural environments.

Our qualitative findings support this interpretation. Women were more likely to describe relational experiences with nature, often linking these to caregiving roles, community involvement, and environmental activism. These reflections were consistent with a search for meaning that is grounded in action and connection. In contrast, men more frequently referenced theological or textual sources—such as Tikkun Olam, Torah commandments, and structured rituals like Shabbat—as frameworks for environmental responsibility and moral clarity. Their reflections conveyed a sense of presence of meaning rooted in tradition and communal identity.

Bronfenbrenner’s Ecological Systems Theory offers a layered framework for understanding how individuals derive meaning through their relationship with nature, particularly in the face of climate change. At the macrosystem level, broad cultural ideologies, religious narratives, and societal norms shape environmental attitudes and moral orientations. For individuals affiliated with the Progressive Jewish Movement, values such as Tikkun Olam—understood as the ethical imperative to “repair the world”—are central. This concept, while often seen today as a call for social and environmental justice, has deep historical roots: it originates in rabbinic literature, was further developed in the 16th century by Isaac Luria within the Kabbalistic tradition, and later gained prominence in modern liberal Jewish thought. Tikkun Olam has evolved into a guiding principle for ethical action, social responsibility, and ecological care ([12]). This evolution illustrates how religious ideals adapt to contemporary environmental and social challenges, embedding ecological concern within a larger cultural and moral framework ([25]). Such macrosystem-level ideologies supply moral architecture through which environmental behavior is rendered meaningful.

Social Role Theory complements this view by suggesting that cultural narratives intersect with gender norms: women may frame environmental concern through a relational and care-oriented lens, while men may emphasize duty, continuity, and moral leadership. These patterns align with Relational-Cultural Theory, which posits that women more often derive meaning through mutual connection, including relationships with the natural world ([24]; [29]). If nature is experienced as a relational entity, then women may engage with it in ways that mirror their approach to human relationships, strengthening the link between nature relatedness (NR) and meaning in life (MIL).

At the microsystem level, individuals’ daily lives within families, congregations, and communities provide embodied expressions of these values. Jewish holidays such as Sukkot, Pesach, and Shavuot—though rooted in historical and religious tradition—also follow the agricultural calendar, highlighting humanity’s dependence on the Earth and encouraging ecological mindfulness ([42]; [6]). These rituals reinforce a sense of continuity and purpose amid environmental disruption, anchoring people’s experience of climate stress in communal, spiritual rhythms. Finally, broader cultural narratives—such as those linking progressive ideological frameworks with environmental concern—further support this integration of ethics, identity, and ecological responsibility ([31]). Together, the microsystem and macrosystem illustrate how meaning-making around nature is shaped by both intimate relational contexts and broader sociocultural ideologies, especially for members of a religious minority whose traditions offer resilience, continuity, and moral clarity in the face of ecological uncertainty.

## 5. Conclusions

This study contributes to our understanding of how NR and MIL interact within a culturally specific, gendered, and ecologically challenged context. Our findings indicate that while NR may not guarantee a stable presence of meaning, it significantly contributes to the active search for meaning, especially in the face of environmental disruption. This supports [16]’s ([16]) view of meaning as a striving process rather than a static state. Gender emerged as a meaningful lens through which NR is experienced and translated into meaning. Quantitative and qualitative results jointly revealed that women’s relationship with nature was often framed through relational and action-oriented terms, while men more commonly articulated meaning through textual traditions and collective identity. These patterns align with Relational-Cultural Theory and Social Role Theory, which together illuminate how gendered socialization shapes not only behavior but also existential orientation.

The Jewish Progressive context added further richness, demonstrating how religious identity can mediate and amplify the NR–MIL link through cultural rituals, moral narratives, and evolving theological concepts such as Tikkun Olam. Jewish festivals that reflect seasonal cycles provide a framework for environmental concern and meaning-making. Particularly in times of ecological uncertainty, these traditions offer continuity, coherence, and moral grounding.

### 5.1. Limitations and Future Research

Together with its valuable insights, this study has several limitations that should be acknowledged. First, the cross-sectional design prevents drawing causal inferences about the relationship between NR and MIL. Although significant associations were found, their direction remains unclear; future longitudinal or experimental work is essential to clarify causal pathways. Second, the sample consisted mainly of prominent Jewish community leaders in Australia—rabbis, educators, and lay leaders. While their perspectives are rich and influential, they do not capture the full diversity of the Jewish population. Subsequent studies should recruit participants with different levels of religious observance, socioeconomic status, and community involvement. Including voices outside leadership positions would provide a more inclusive and nuanced account of how NR and MIL operate across subgroups. Third, the association between gender and NR was only marginally significant, so replication with larger, more diverse samples is needed to confirm its robustness. Another limitation is that the qualitative questions were not fully open-ended, which may have limited participants’ ability to express how, if at all, Jewish traditions shaped their relationship with nature. Finally, data were gathered between November and December 2023, shortly after the Hamas attacks and during Israel’s Iron Sword war—events that may have influenced participants’ emotional states and responses, particularly regarding meaning, identity, and connection to nature.

### 5.2. Implications for Practice and Community Engagement

Women in our sample tended to turn religious principles into hands-on environmental projects, whereas men emphasized theoretical or communal discussions of stewardship. Programs that align with these preferences—e.g., women-led gardening workshops and men-led study groups on Jewish ecological law—can harness the distinct motivations of each group. Designing educational campaigns and sustainability projects around these gendered strengths can boost overall participation. Community committees can combine practical initiatives—like recycling drives and community gardens—with policy briefings or text-study sessions, creating a well-rounded, community-wide response to environmental challenges.

Meaning derived from contact with nature is a stronger driver of pro-environmental behavior than life satisfaction alone ([10]). Embedding NR in community programs and psychotherapy can serve two complementary goals: strengthening mental health and encouraging sustainable action. Framing environmental loss as an ethical imperative can further build psychological resilience in the face of climate change. In clinical practice, adding NR to meaning-centered therapy is an optional way to ease existential distress. Simple nature-based activities—such as mindful walks or quiet outdoor reflection—can lower stress and help clients link their sense of purpose to the natural world.

## Figures and Tables

**Figure 1 behavsci-15-01045-f001:**
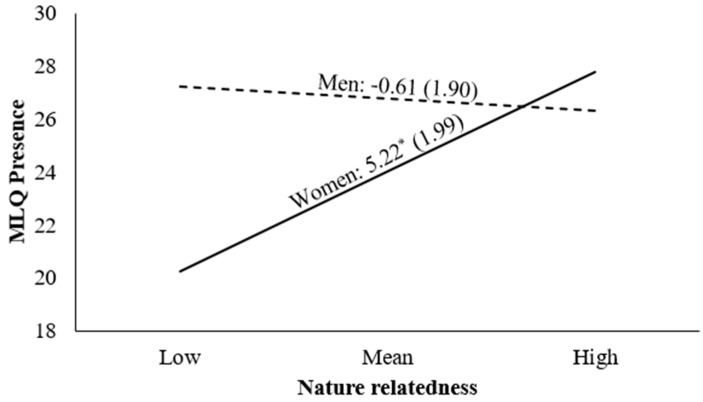
Simple slopes technique for plotting the interaction between nature relatedness and gender on MLQ Presence. *Note. N* = 35. Values are unstandardized regression coefficients. In parentheses: standard errors. MLQ = Meaning in Life Questionnaire. * *p* < 0.05.

**Table 1 behavsci-15-01045-t001:** Means, standard deviations, and intercorrelation between NR and MLQ measures, and their associations with background variables.

Variable	1	2	3
Nature Relatedness	–		
MLQ Search	0.39 *	–	
MLQ Presence	0.18	0.14	–
Gender ^a^	0.39 ^†^	0.03	−0.16
Age	0.02	−0.18	0.09
Birthplace ^b^	0.05	0.18	−0.18
Years of education	0.17	−0.06	−0.01
Family status ^c^	−0.21	−0.15	0.01
Number of children	−0.03	−0.11	0.06
Economic status ^d^	−0.27	−0.26	−0.10
*M*	3.88	20.66	25.77
*SD*	0.72	7.81	5.87

*Note*. *N* = 35. MLQ = Meaning in Life Questionnaire. ^a^ 0 = Men, 1 = Women; ^b^ 0 = Other, 1 = Australia; ^c^ 0 = Not in a relationship, 1 = In a relationship; ^d^ 0 = Doing well, 1 = Doing very well. * *p* < 0.05. ^†^
*p* < 0.10.

**Table 2 behavsci-15-01045-t002:** Results of hierarchical multiple regression analyses predicting MLQ measures.

	MLQ Search			MLQ Presence		
Variable	*B*	*SE B*	β	*B*	*SE B*	β
Step 1						
Gender ^a^	−1.67	2.69	−0.11	−2.95	2.11	−0.25
Nature Relatedness	4.57	1.85	0.42 *	2.16	1.45	0.27
	*R*^2^ = 0.16 ^†^			*R*^2^ = 0.09		
Step 2						
Gender ^a^	−1.71	2.73	−0.11	−2.74	2.00	−0.23
Nature Relatedness	5.11	2.59	0.42 ^†^	−0.61	1.90	−0.08
Nature Relatedness × Gender ^a^	−1.13	3.76	−0.07	5.83	2.75	0.48 *
	*R*^2^ = 0.16			*R*^2^ = 0.21 ^†^		
	Δ*R*^2^ = 0.002			Δ*R*^2^ = 0.12 *		

*Note*. *N* = 35. MLQ = Meaning in Life Questionnaire. ^a^ 0 = Men, 1 = Women. * *p* < 0.05. ^†^
*p* < 0.10.

**Table 3 behavsci-15-01045-t003:** Analysis by gender and self-reflection on Jewish traditions and practices towards NR and MIL.

Analyses of Male Participants			
Initials (Age)	State	Description	Statement
JF (85)	Victoria	Born in Austria and immigrated to Australia after studying medicine in the USA. Married with three daughters. He is a secular Jew and is not affiliated with any specific Jewish congregation.	I do not see Jewish traditions as being linked to any environmental issues.
PB (78)	Victoria	Originally from England, immigrated to Australia in his twenties, and has since made Melbourne his home, where he worked as a physician. Married with four children. Active member of the Progressive Movement.	The practice of “Bal Tashlich”, symbolically gathering and removing debris from natural spaces, is “a meaningful cultural method to care for the planet”.
CE (77)	Victoria	Born in England and immigrated to Australia in his twenties. Worked in the academic domain, researching plant biology. Married with two sons, he is an active member in the Jewish Progressive Movement.	“If you live a Jewish life, you must care about the environment. We must show concern and not be destructive toward it. The fundamental core of faith is connected to the concept that we are part of nature, that we neither possess nor control it. Judaism teaches this”.
DK (76)	Victoria	Born and raised in England, immigrated to Australia after marriage. Married with four children and worked as an accountant. Now retired and active in the Progressive Movement.	“The Torah teaches all about how to change from a nomadic to an agrarian community. Every day, we acknowledge in the Amidah prayer the importance of sun, rain, dew and the need for a benign climate to revive, replenish and recreate”.
VR (70)	Victoria	Born and raised in Melbourne, married with three children, works as a physician.	Attends synagogue only on Rosh Hashanah and Yom Kippur. Takes pride in his Jewish heritage but does not explicitly link it to environmental activism.
DA (69)	Victoria	Born and raised in Melbourne, pursued a career in education. Following a divorce ten years ago, actively involved in environmental organizations and the progressive Jewish community.	“Jewish teachings are consistent with and support my principles and practices, such as *Tikkun Olam*”.
RA (53)	Victoria	Born in the USA, immigrated to Australia three decades ago. Married with four children, works as an educator, active in the progressive Jewish community.	“The Torah teaches that humanity was created to be caretakers/gardeners of the earth. Plant and animal life was created before humanity. We are all part of God’s creations. Various mitzvot about how to treat the land and animals ethically demonstrate the Jewish mandate to care for all of God’s creations”.
JG (66)	New South Wales	Emigrated from South Africa to Australia in his twenties, works as a financial advisor. Does not belong to a specific Jewish community.	Skeptical that *Tikkun Olam* is directly related to climate change.
GB (68)	Western Australia	Originally from the USA, settled in Western Australia in his twenties. Works in agriculture. Married with three children.	“Local communities have the power to create meaningful environmental change”.
FP (58)	Western Australia	Born in France, immigrated to Australia as a teenager with his parents. Works as an architect, participates weekly in Progressive Jewish Movement activities.	“Jewish teachings stress respect for nature, other humans, and animals”.
AK (69)	Western Australia	Married with three children, born in the USA, moved to Perth following his Australian wife. Retired after working as an economic consultant for a high-tech company. Active in the Progressive Movement.	“We do have a responsibility for *Tikkun Olam*, which means if we are doing something wrong in relation to the world we live in, we should do more to stop causing the problem and reverse the damage”. He elaborates: “Having a Shabbat meal with the family on Friday nights seems to be helpful for reflecting on the issues of the world, by taking a step back and not being so caught up in day-to-day issues. Lighting candles seems particularly soothing”.
**Analyses of Female Participants**			
**Initials (Age)**	**State**	**Description**	**Statement**
SB (75)	Victoria	Born and raised in Melbourne, divorced with one adult daughter. Previously employed in journalism, now dedicates time to volunteering with children and attending progressive community activities.	“Judaism’s emphasis on the sanctity of human life and resilience encourages persistence in taking care of our planet”. She cites the story of Noah as an instructive narrative for environmental responsibility and suggests that rabbinic commentary can further illuminate these teachings.
ME (74)	Victoria	Born and raised in Melbourne, widowed for a decade, living near her only son. Former educator, now volunteers and is active in the progressive Jewish community.	No specific environmental viewpoint articulated.
PD (74)	Victoria	Born and raised in Victoria, Australia. Married and spent many years working in education. Now retired and devotes time to volunteering within the progressive Jewish community.	“Many Jewish holidays are tied to the seasons, agriculture, and the land. These traditions remind me to remain vigilant in protecting resources and ensuring a healthier planet for future generations”.
EL (72)	Victoria	Born and raised in Melbourne, married with two adult children. After a career as an educator, now volunteers with environmental organizations and participates in the Progressive Movement.	“Jewish teachings can show people how to live and take care of the land and animals”.
LL (71)	Victoria	Born and raised in Victoria, married with one adult daughter, identifies as a secular Jew.	“I participate in protests, and cycle as a form of eco-friendly transport”.
MA (65)	Victoria	Born and raised in southern Victoria, divorced with two children. Retired teacher and environmental activist.	“Humanity shares the Earth with countless other organisms and must act as good stewards”. Sees environmental care as a key Jewish value.
YA (63)	New South Wales	Born and raised in New South Wales, moved to southern Victoria. Married with two daughters. Homemaker, volunteers once a week with animals, identifies as a secular Jew.	“Jewish traditions point to the interconnectedness of the world, and thus the need to be aware of other species and the flow of the world”.
ZB (60)	Victoria	Born and raised in Melbourne, divorced with three children. Secular Jew, not affiliated with any Jewish congregation.	“Jewish traditions point to the interconnectedness of the world, and thus the need to be aware of other species and the flow of the world”.
JT (59)	Victoria	Born and raised in Melbourne, married with three children, volunteers with various art organizations.	“Jewish teachings are relevant in relation to animals, where one always ensures the animals are looked after and fed before oneself; we must look after those that cannot look after themselves”.
CB (40)	New South Wales	Single mother of one daughter, works as a family consultant in Sydney. Participates sporadically in local Progressive Movement events.	“Jewish teachings can foster greater environmental education and informed decision-making about climate issues”.
AH (22)	Victoria	Born and raised in Melbourne, literature student, identifies as a secular Jew, not affiliated with any Jewish congregation.	“[I try] to make changes in my life that are good for the environment”.

## Data Availability

The datasets of the current study are available from the corresponding author upon reasonable request.

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
