# Peer review of "Climate Crisis and Mental Well-Being: Nature Relatedness, Meaning in Life, and Gender Differences in a Jewish Australian Study"

_behavsci, 2025, doi:10.3390/bs15081045_

Round 1
Reviewer 1 Report
Comments and Suggestions for Authors
Thank you for the opportunity to review the article. The article is interesting and important, and I thoroughly enjoyed reading it. It addresses a timely and significant topic, especially in the current context of climate crisis and mental well-being. Furthermore, the exploration of how people find meaning in life through their connection to nature, along with the cultural and gendered context, adds depth and originality to the article.
The article is well-written and very clear. The literature review is detailed and coherent, effectively guiding the reader toward the research objectives.
I would like to offer a few suggestions for improvement:
-
Theoretical framework: It may be beneficial to consider referencing or incorporating ecological theory, which is highly relevant to the conceptual framework of the study. This could add depth to the use of social role theory.
-
Methodology: The combination of quantitative and qualitative traditions enriches the research. However, the phrasing of the qualitative questions does not align well with the qualitative approach. It would have been preferable to use open-ended questions and ask participants to describe the Jewish traditions that shaped (if at all) their relationship with nature. Since the questions were not phrased openly, I suggest including this as a limitation of the study.
-
Qualitative data analysis: Please elaborate on the stages of analysis, provide examples for each stage, and describe how the categories and themes were formed. How many researchers were involved in the analysis? How was agreement reached?
-
Ethical considerations: The article lacks a section describing the ethical aspects of the study.
Author Response
Dear editor,
I am deeply grateful to the reviewers for their constructive comments. I have addressed all of them in this version of the manuscript. I have also added several references and included them all in the reference list.
Reviewer 1
Thank you for the opportunity to review the article. The article is interesting and important, and I thoroughly enjoyed reading it. It addresses a timely and significant topic, especially in the current context of climate crisis and mental well-being. Furthermore, the exploration of how people find meaning in life through their connection to nature, along with the cultural and gendered context, adds depth and originality to the article.
The article is well-written and very clear. The literature review is detailed and coherent, effectively guiding the reader toward the research objectives.
I would like to offer a few suggestions for improvement:
- Theoretical framework: It may be beneficial to consider referencing or incorporating ecological theory, which is highly relevant to the conceptual framework of the study. This could add depth to the use of social role theory.
My response: Thank you. Bronfenbrenner’s Ecological Systems Theory (1979) was incorporated to structure the theoretical framework and explore how Nature Relatedness (NR) and Meaning in Life (MIL) are shaped across nested environmental systems. Each system level was examined in relation to gendered developmental processes and cultural positioning. Lines 107-149.
Furthermore, Bronfenbrenner’s model provided an organizing lens in the Discussion section. Lines 431-469.
- Methodology: The combination of quantitative and qualitative traditions enriches the research. However, the phrasing of the qualitative questions does not align well with the qualitative approach. It would have been preferable to use open-ended questions and ask participants to describe the Jewish traditions that shaped (if at all) their relationship with nature. Since the questions were not phrased openly, I suggest including this as a limitation of the study.
My response: I added it as a limitation. Lines 503-505.
- Qualitative data analysis: Please elaborate on the stages of analysis, provide examples for each stage, and describe how the categories and themes were formed. How many researchers were involved in the analysis? How was agreement reached?
My response: Stages of thematic analysis aligned with Braun & Clarke (2006; 2021) are now added. Lines 268-297.
- Ethical considerations: The article lacks a section describing the ethical aspects of the study.
My response: Added. Lines 164-168.
Thank you for the opportunity to make the revisions and for granting me the extension for the submission.
I truly appreciate it.
Sincerely
Prof. Orly Sarid
Reviewer 2 Report
Comments and Suggestions for Authors
Thank you for the opportunity to review this interesting article. It is a well written piece that provides additional information into the impacts of nature and wellbeing. I have a few suggestions which I have outlined below. I hope you find these useful.
Abstract:
- Need to state what acronyms stand for prior to using them
Introduction
- Line 41-43- The definitive statement used here makes it sound as though Clayton et al, 2021 evaluated the psychological impacts of the changes to Australian climate specifically- which is not the case. Therefore this should be reworded to state ‘which may contribute to..’ or something similar.
- 44-46 - This sentence needs rewording for clarity
- Line 50- need to explain what the acronyms MIL and NR represent before using these as standalone acronyms
- Throughout the introduction the terms ‘positive mental health, psychological wellbeing, and psychological health are used interchangeably which is confusing for the reader. Choose one term and explain early on in the introduction which term will be used and why
- 129-134- formatting error where the ‘We hypothesize that’ should not be numbered
Methods
- Need to state how ethics was obtained for this study
- Line 141- why were participants over the age of 21- As legal age in Australia is 18 this is the typical cut off used for research, so need to justify why 21 was chosen instead
Procedure
- Line 172- restate how the participants were approached (e.g. via email/social media)
- Line 174- was the questionnaire completed digitally or on paper?
Data analysis
- Line 216- needs to state ‘data were’ not ‘data was’ as data is plural
- Line 231- Need to include year of publication for Braun & Clarke- especially as they have different versions of their framework
Conclusion:
- Would like to see a broad conclusion at the end of the discussion to summarise the research.
Author Response
Dear editor,
I am deeply grateful to the reviewers for their constructive comments. I have addressed all of them in this version of the manuscript. I have also added several references and included them all in the reference list.
Reviewer 2
Thank you for the opportunity to review this interesting article. It is a well written piece that provides additional information into the impacts of nature and wellbeing. I have a few suggestions which I have outlined below. I hope you find these useful.
Abstract:
Need to state what acronyms stand for prior to using them
My response: Done. Line 12.
Introduction
Line 41-43- The definitive statement used here makes it sound as though Clayton et al, 2021 evaluated the psychological impacts of the changes to Australian climate specifically- which is not the case. Therefore this should be reworded to state ‘which may contribute to..’ or something similar.
My response: I reworded the sentence. Lines 41-44.
44-46 - This sentence needs rewording for clarity.
My response: I reworded the sentence. Lines 45-47.
Line 50- need to explain what the acronyms MIL and NR represent before using these as standalone acronyms
My response: Done.
Throughout the introduction the terms ‘positive mental health, psychological wellbeing, and psychological health are used interchangeably which is confusing for the reader. Choose one term and explain early on in the introduction which term will be used and why
My response: To maintain clarity and consistency, this paper adopts the single term “positive mental health” to denote optimal psychological functioning and subjective wellbeing (Keyes, 2005). Added in lines 63-65,
129-134- formatting error where the ‘We hypothesize that’ should not be numbered
My response: Corrected.
Methods
Need to state how ethics was obtained for this study
My response: Added. Added. Lines 164-168.
Line 141- why were participants over the age of 21- As legal age in Australia is 18 this is the typical cut off used for research, so need to justify why 21 was chosen instead.
My response: Added. Lines 171-178.
Procedure
Line 172- restate how the participants were approached (e.g. via email/social media)
My response: Added. Line 210.
Line 174- was the questionnaire completed digitally or on paper?
My response: Digitally. Added. Line 211.
Data analysis
Line 216- needs to state ‘data were’ not ‘data was’ as data is plural
My response: Corrected.
Line 231- Need to include year of publication for Braun & Clarke- especially as they have different versions of their framework
My response: Thank you. Done.
Would like to see a broad conclusion at the end of the discussion to summarise the research.
My response: A conclusion section is now added after discussion before limitations. Lines 471-488.
Thank you for the opportunity to make the revisions andfor granting me the extension for the submission.
I truly appreciate it.
Sincerely
Prof. Orly Sarid